# Improving Learning and Study Strategies in Undergraduate Medical Students: A Pre-Post Study

**DOI:** 10.3390/healthcare11030375

**Published:** 2023-01-28

**Authors:** Ivan Sisa, María Sol Garcés, Cristina Crespo-Andrade, Claudia Tobar

**Affiliations:** 1Escuela de Medicina, Colegio de Ciencias de la Salud, Universidad San Francisco de Quito USFQ, Quito 170901, Ecuador; 2Instituto de Enseñanza y Aprendizaje IDEA, y Academia SHIFT, Colegio de Ciencias Sociales y Humanidades, Universidad San Francisco de Quito USFQ, Quito 170901, Ecuador; 3Instituto de Neurociencias, Universidad San Francisco de Quito USFQ, Quito 170901, Ecuador

**Keywords:** learning and study strategies, study habits, Latin America

## Abstract

We aimed to describe the impact of a structured interventional program to improve learning and study skills in undergraduate medical students from a Latin American medical school. The interventional program’s design was based on diagnostic/prescriptive assessment test scores measuring ten scales. The program consisted of five tailored workshops. The cohort studied consisted of 81 third-year medical students. The outcome variable was the difference between “pre” and “post” test scores. The unadjusted score percentiles were used to compare improvement in learning and study skills. In addition, a sensitivity analysis was conducted to assess variation in the mean difference of the test scores by the number of workshops attended. The response rate was 100% (81/81) for the pre test and ~77% (62/81) for the post test. After the interventional program, nine out of ten scales showed statistical improvement, except for the scale of motivation. The scales with the highest and lowest percent change improvement were time management (66%, *p*-value: <0.001) and motivation (14.9%, *p*-value: 0.06). The students who attended more workshops obtained a higher percent change improvement in the post test. These findings suggest that through a well-designed interventional program, it is possible to improve learning and study skills among medical students.

## 1. Introduction

Medical education literature reveals that a large population of undergraduate health sciences students lack good study skills and habits [1,2]. Thus, introducing a study skills program could significantly improve student confidence and academic performance [2]. Medical education researchers have implemented several methods to identify students at risk of academic failure during medical school [3,4]. However, despite these efforts, most of those students will be identified far too late in their academic preparation. Therefore, many at-risk students with undiagnosed study and learning difficulties will be unnoticed from the start without receiving the appropriate support they need to succeed [3]. It is imperative to identify those students as soon as possible; otherwise, they are more likely to adopt inappropriate learning strategies and use maladaptive coping styles against failure, and less likely to participate in academic support programs [5]. This scenario has been associated with higher prevalence of depression and anxiety among medical students compared to the general population [6], which ultimately increases the odds of medical student dropout [7]. Medical school attrition has significant consequences not only for the academic institution itself but also for the society as a whole [8]. This phenomenon is even more pervasive in medical schools based in low-and middle-income countries (LMICs) whose dropout rate is between 40 and 63% compared to the global average of 11.1% [9,10]. Some factors associated with medical school dropout include socio-demographic problems, lower entry qualifications, psychological attributes and signs of academic struggle during medical school coursework, this last factor being strongly associated with dropout [7].

Most of the evidence regarding identifying at-risk students and potential interventional programs to help those students comes from the USA and countries in Europe such as the UK and the Netherlands [3,4,7,8,11,12]. The Learning and Study Strategies Inventory (LASSI) test is one of the most common diagnostic and prescriptive assessments used to provide early information regarding the strengths and weaknesses of how students learn and study [11]. Overall, the LASSI test has shown the appropriate psychometric characteristics that would allow it to be used as a diagnostic and prescriptive tool for undergraduate students, even those living in a Spanish speaker country [13,14]. Yet, while most of the studies using the LASSI test have evaluated its correlation with student performance, they have not proposed a structured intervention to help the medical students identified as “at-risk” [11,15,16,17]. Few studies enrolling non-medical students have demonstrated the utility of LASSI scores with improved performance after an intervention [18,19]. The aim of this study was to design, implement, and measure the impact of a structured interventional program named “USFQ’s Keys to Academic Success Program” (UKASP). The program was designed based on LASSI scores of third-year medical students from a Latin American medical school, and intended to improve learning and study skills. To inform medical education literature in this regard, we conducted a “pre-post” study comparing LASSI scores before and after the designed program.

## 2. Materials and Methods

### 2.1. Study Design and Theoretical Framework

We conducted a pre-post study to evaluate the impact of a structured interventional program of five workshops. To guide our tailored interventional program, we applied the Model of Strategic Learning (MSL) developed by Weinstein [20]. This emergent model states that successful learning emerges from the interactions among a student’s skill, will, and self-regulation and cannot be attributed to a single component. “Skill” refers to behavioral and cognitive processes related to the construction of meaning. “Will” includes learning attitudes, motivation, and anxiety concerning academic performance. “Self-regulation” refers to the management of the learning process by planning, monitoring, focusing, and evaluating knowledge and skills [20,21]. MSL highlights that learning and studying are active processes under the control of students that can be proactively and intentionally used to improve learning and academic success [20]. Teaching students learning strategies involves direct instruction, modeling, and guided practice with feedback. Thus, Weinstein et al. developed the LASSI test to diagnose the strengths and weaknesses of students in relation to the above components [20,22].

### 2.2. Study Setting

We conducted this study at the Universidad San Francisco de Quito (USFQ) School of Medicine located in Quito, Ecuador. The USFQ School of Medicine uses a 6-year curriculum to train high school graduates; in Ecuador, as in many other LMICs, it is not a requirement to have a previous degree to apply and be accepted into medical school [23]. Its curriculum is arranged with the first two years dedicated to study of basic sciences and a liberal arts education which aims to foster and develop critical thinking, creativity, and research attitude. During the third year (preclinical transition year) students are challenged to use their knowledge in the study and resolution of clinical cases. Also, during this year students develop abilities and skills to examine high-fidelity mannequins and standardized patients. The last three years are devoted to clinical sciences training, community work, and hospital internship [24]. Approval for this study was obtained from the ethics committee of USFQ (#2019-113IN). This study was executed according to the Declaration of Helsinki.

### 2.3. Study Sample

The present study invited all third-year medical students (n = 107) registered during the academic year 2019–2020 at USFQ School of Medicine to participate. We chose third-year students because we aimed to use the National Board of Medical Examiners Comprehensive Basic Science Examination (NBME CBSE) scores as outcome data and consider its association with the implemented workshops. The USFQ School of Medicine uses the NBME CBSE progress test starting in the program’s third year. The inclusion criteria to participate in the study were: (i) having signed the informed consent, (ii) being a full-time student at the USFQ School of Medicine, and (iii) having completed the pre-intervention LASSI test. A total of 81 participants consented to participate in the study.

### 2.4. Measures

We collected baseline characteristics (sociodemographic, educational, and behavioral) of the study participants using a survey distributed by REDCap. This survey was built by the research team members with the support of an expert in Survey Research and Methodology who reviewed and critiqued the questions before shared by REDCap. To diagnose learning and study skills weaknesses among the study’s participants, we used the LASSI test to assess students’ awareness of learning and study strategies related to the skill, will, and self-regulation components of strategic learning. The inventory is a self-report instrument with 60 items divided into 10-scales arranged in three components as follows [25]:(i).*Skill component*: information processing (INP), selecting main ideas (SMI), and test strategies (TST);(ii).*Will component*: anxiety (ANX), attitude (ATT), and motivation (MOT);(iii).*Self-regulation component*: concentration (CON), self-testing (SFT), study aids (STA), and time management (TMT).

The LASSI test is a computer-based test; after the user has completed all the items using a five-point Likert scale (Not at all typical of me, Not very typical of me, Somewhat typical of me, Fairly typical of me, and Very much typical of me) and successfully submitted a two-page report, the program will display a list of the scores for each of the ten scales. As LASSI is a diagnostic instrument, no total score is computed. Instead, it provides standardized percentile scores on an individual scale. The scores obtained can then be compared to the norms provided, to local norms, or to cut-off scores developed by an institution or program [22]. According to the developer, a student with a score close to or below the 50th percentile on a particular scale should seek help to improve the related skills and succeed during college education [11,24,25]. Furthermore, students scoring from the 50th to the 75th percentile of a particular scale should consider improving that skill or strategy, and students scoring above the 75th percentile in any scale typically do not need to improve [11].

### 2.5. Intervention

In line with the MSL, and based on the pre LASSI test scores, we designed and tailored the UKASP. The intervention consisted of five workshops covering the eight scales that received the lowest average score among the study group (Figure 1A). In each workshop, two main scales were addressed, and related elements of the other scales were incorporated if they had relevance. Note that the two lowest scales (TMT & ATT) were included twice throughout the workshops for reinforcement (Figure 1B).

Each workshop had a duration of 90 min and was conceptualized and delivered by a multidisciplinary team (composed of a medical educator, two psychologists, and one educator), rather than only medical school faculty members. The workshops’ structure was designed based on Kolb’s Learning Cycle, since this instructional model provides a blend of traditional teaching and hands-on learning. Following this model, the workshops’ structure included a four-stage experiential learning cycle as follows: concrete experience, reflective observation, abstract conceptualization, and planning active experimentation [26]. The first workshop, focusing on attitude and motivation, aimed to challenge students’ mindsets, establish motivational factors, and reflect on values and goals. In this session, students played a “true vs. false” game related to goals, values, and motivation theory; worked in groups on a mindset card exercise where they had to organize similar ideas into categories; and reflected on which group of cards represented their way of thinking. Additionally, they learned about mindset and motivation theory. Finally, they applied a value identification exercise tool called the “Value COMPASS” that helps students review and reflect on their values and personal goals [27].The second workshop addressed students’ time management and attitude toward studying and achievement outcomes. In this session, students completed the Jar of Life Experiences Exercise [28], which illustrates time management principles; they reflected on expectation vs. reality time management in their academic lives and learned about prioritization, reflection, and the SMART Goals Model. SMART is the acronym for Specific, Measurable, Achievable, Relevant, and Time-bound [29]. Finally, they completed a SMART Goal Template, a helpful tool to guide and improve goal identification and prioritization, and a goal achievement plan.

The third workshop focused on selecting main ideas and test strategies, reviewing strategies to identify relevant information from academic sources, reasoning to answer questions, and preparing for tests. Students watched two videos from “Brain Games” related to attention illusions, reflected on their study habits, learned how cognitive functions work, and practiced identifying main ideas identification with a reading exercise and a memory experiment during the workshop. The fourth workshop focused on anxiety and concentration, and aimed to identify academic factors that trigger anxiety and prompt participants to reflect on their anxiety levels, learn techniques for coping with anxiety, and focus on their academic tasks [22]. During this session, students watched a fragment of the Angst documentary designed to raise awareness about anxiety [30]. Then they answered The Holmes-Rahe Stress Inventory [31], which measures an individual’s exposure to environmental stress. In addition, they learned how college students experience stress with data and completed a Strategies Checklist of self-care ideas.

The fifth workshop addressed students’ time management and the use of academic sources. It aimed to increase awareness of their time management and knowledge of academic resources to avoid procrastination. The session started with the students presenting the learning strategies that they had used, and describing which ones had worked for them. Then students categorized these strategies into highly effective, moderately effective, and non-effective strategies. Next, they reflected upon their strategies to study when they did not feel like studying. Then they learned about time management techniques, procrastination, and intrinsic motivation. Finally, they practiced the Pomodoro Technique, which provides a tool for improving productivity, enhancing focus and concentration by cutting down on interruptions, and alleviating the anxiety linked to time management [32].

Table 1 shows in detail the program and the components of each planned workshop and the LASSI scales addressed during each session.

## 3. Procedure

As part of an internal Grant Program, USFQ School of Medicine funded the UKASP to assess the impact of five face-to-face workshops to strengthen learning and study strategies in third-year medical students. The program started in the fall semester of 2019–2020. During this semester, administrative (i.e., IRB application) and logistics issues were resolved. The pre-LASSI test was applied in December 2019; the workshop interventions started in February 2020; and we delivered the final workshop in December 2020. Due to the COVID-19 pandemic, we had to migrate from face-to-face to virtual format, via Zoom (online video conference software), after the second workshop. After the fifth workshop, a post-LASSI test was conducted to evaluate the impact of the UKASP (Appendix A). The DoCTRINE (Defined Criteria to Report Innovations in Education) guidelines were used to ensure the proper reporting of this educational innovation study [33].

### 3.1. Statistical Analysis

Using an online calculator for a pre-post study, it was estimated that a sample size of 34 participants was necessary to assess an effect size of 0.5 with an alpha error of 5%, power of 80%, and using a conservative standard deviation of 1 [34]. We used descriptive statistics to summarize the baseline characteristics of the study participants. We described continuous variables as mean ± SD and categorical variables as counts and percentages. Since we used one-group pre-post study design with matching scores, the paired-sample t-test was used to assess the difference between pre- and post-LASSI test scores [35]. In addition, we performed a sensitivity analysis to assess variation in the differences in the means of the pre- and post-LASSI test scores by the number of workshops attended. We considered a two-tailed *p*-value < 0.05 sufficient to indicate statistical significance. All data management and analysis was conducted using RStudio v.1.1.463 for macOS.

### 3.2. Results

The final study sample for this pre and post study consisted of 81 students (75.7%, 81/107). Seventy-eight students answered the REDCap survey (96.3%, 78/81) regarding baseline characteristics. The mean age of the study participants was 21.4 (±1.2) years, with 65% being female, 3.8% admitted to the medical school with a previous bachelor’s degree, 60.2% with financial aid to support their medical education, and more than 9% with a chronic or mental disease (Table 2). In addition, despite not finding demographic differences between study participants who completed the UKASP compared to those who did not, we did find differences in the pre-LASSI test scores (Appendix A). The response rate was 100% (81/81) for the pre-LASSI test and ~77% (62/81) for the post-LASSI test. Therefore, 62 students made up the rest of the analysis. Figure 2 shows the aggregate data results obtained in the pre- and post-LASSI tests.

Overall, the group scored less than the 50th percentile in almost all of the ten LASSI scales, except for the INP and SFT scales. After the workshop interventions, the ATT, CON and STA scales did not surpass the 50th percentile cut-off value, and the scale of TMT obtained a borderline score. However, in the univariate analysis, we noticed that the interventional program had a positive impact on the ten LASSI scales (*p* < 0.05) except for the MOT scale (Table 3).

The LASSI scales with the highest and lowest improvement were TMT and MOT, with 66% and 14.9%, respectively. Due to the COVID-19 pandemic and the voluntary nature of participation, we had a decrement in the workshop’s attendance during the synchronous sessions via Zoom. For example, 100% (62/62) attended the first two face-to-face workshops, 41.9% (26/62) attended ≥3 workshops sessions, 12.9% (8/62) attended ≥4 sessions, and only 6.4% (4/62) attended 5 workshops. As with the primary analysis, the sensitivity analysis showed a positive and statistically significant impact of the workshop interventions (Figure 3).

Furthermore, a higher percentage improvement was evident in the students who attended three or more workshops compared to only two (Appendix A). It is worth mentioning that students with the lowest pre-LASSI scores overall attended more workshops compared to their counterparts (Appendix A).

## 4. Discussion

This pre-post study examined the effect of a structured interventional program based on a valid and reliable tool to diagnose students’ study skills. The primary findings were as follows: (i) Except for INP and SFT scales, the pre-LASSI scores were less than the 50th percentile; (ii) after the structured interventional program, nine out of ten scales showed statistical improvement (all except MOT); and (iii) students who attended more workshops obtained a higher percent change improvement in the post-LASSI test.

### 4.1. Comparison with Other Studies

A common feeling among incoming medical students, especially among those starting medical school straight from high school, is the sensation of not knowing how to learn or study [36]. As found in this study, the pre-LASSI test scores were lower than the recommended cut-off value of 50th percentile, except for SFT (51.2 ± 27.6) and INP (54.4 ± 27.6) scales. We were surprised by this finding, because our study sample consisted of third-year medical students. However, this same perceived lack of study and learning skills has been documented in other settings that included medical students of first to fourth years and other health-related careers such as dentistry, where the highest mean value obtained was 28.67 ± 4.4 (TST scale) [15,37]. Comparing this performance with medical students who are required to have an undergraduate degree (typically at the bachelor level) to enter medical school, there is a striking difference [17]. For example, Khalil et al., analyzing a cohort of 128 first-year medical students, reported a lowest mean value of 59.8 on the SFT scale and a highest mean value of 77.5 on the MOT scale [17]. This difference could be explained due to the fact that medical students who already have a bachelor’s degree have developed several learning and study strategies to excel across a rigorous pre-medical curriculum [38,39]. In addition, these applicants have had to pass standardized admissions tests (i.e., MCAT exam) to be eligible for medical school in the USA or Canada [40]. Thus, having a previous degree and the experience of preparing for a high-stakes exam make those incoming medical students better equipped to navigate medical school education. Other studies have previously reported improvement of learning and study skills among college and medical students after implementing a structured interventional program [12,18,41]. However, compared to the present study, these implemented academic support programs had had a longer duration. For example, Winston and colleagues implemented a mandatory cognitive skills program for at-risk medical students for 14 weeks [12]. Likewise, Dill et al. evaluated the impact of a learning skills support program offered as a three-credit-hour course over a semester [18]. Our structured interventional program consisted of five sessions carefully designed based on the group needs identified in the pre-LASSI scores; however, the short intervention period could have been the reason why the MOT scale did not reach statistical significance in the post-LASSI test. Additionally, although the ATT, CON and STA scales showed statistical differences in the post LASSI scores; they did not surpass the 50th percentile threshold. A local study analyzing a 5-year period of Ecuadorian medical school graduates found a 40% dropout rate, the main reasons being motivation and personal objective issues [10]. This finding suggests that among Ecuadorian medical school students the MOT and ATT scales have certain preponderance that must be taken into account in future local studies. Our finding regarding higher score outcomes and the number of workshops attended aligns with other published program interventions, suggesting a dose-response effect [12,41,42,43]. It is worth noting that this behavior happened among students with the lowest scores in the pre-LASSI test, highlighting the necessity to focus on those particular students at high risk of failure but with a high motivation to overcome this status.

### 4.2. Strengths and Limitations of the Study

We built our interventional program based on the scores of a validated and well-known diagnostic/prescriptive assessment tool, the LASSI test. Thus, the pre-LASSI scores allowed us to be more efficient in tackling core learning and study skills deficiencies among the study participants. We used easy access and practical tools such as COMPASS, SMART, and Pomodoro techniques. The pre-post study design and sensitivity analysis performed evinced that the implemented interventional program played a role in the observed effect. In addition, a multidisciplinary team (encompassed by a medical educator, two psychologists and one higher education educator), rather than only medical school faculty members, conceptualized and delivered the interventional program. Among limitations, we should mention that due to the COVID-19 pandemic, we could not provide outcome results using a standardized computer exam (NBME CBSE) as initially planned, and we lost 23.5% (n = 19) of the study participants who did not take the post LASSI test. Despite the fact that we found no differences between the two groups in demographic variables, we cannot rule out the possibility of a confounding effect. Second, compared to other interventional programs, ours implemented a reduced number of interventions (five in total). However, Stegers-Jager et al. conducted a randomized controlled trial to deliver a short-integrated study skills program (five sessions) to support at-risk first-year medical students. This study found positive results among those who attended the short-integrated study skills program compared to the control group [41]. Like Stegers-Jager et al., we found very encouraging the results confirming the belief that approaches to studying and learning can be taught [12,41,44]. A possible explanation for this might be that we built our interventional program based on previous diagnostic information (preLASSI test scores) that could have led to a targeted and more efficient intervention. This approach could save time and resources in planning future similar interventional programs and deserves further exploration. Third, we did not evaluate the perception of the delivered intervention and its impact upon the study participants, and whether this translates to better academic or retention outcomes. Fourth, we cannot exclude the possibility of an effect on or modification of our results due to the COVID-19 pandemic, baseline characteristics (e.g., were some students receiving financial assistantships that would have benefited them more than others?), confounding factors, and even normal student activities such as classes they were taking or studying they were completing. Future experimental investigations in the field should take into account these issues. Fifth, we had considerable dropping-out during the Zoom workshop sessions, which could have attenuated the mean difference in the post-LASSI test. However, due to these factors, the results should be interpreted with caution. Notwithstanding these limitations, it is encouraging that our findings are consistent with earlier observations published elsewhere [12,15,18,38,42,43].

### 4.3. Medical Education Implications

Our study has important implications for medical schools that accept high school graduates and medical education settings with high attrition rates. In those scenarios, medical schools are more likely to receive a very diverse range of students, including first-generation college students [45]. This could explain the higher dropout rate among undergraduate medical students, especially those based in LMICs, compared to the global average, 42–63% vs. 11.1%, respectively [8,9]. Hence, we favor the approach of providing the necessary support to at-risk students in order to adapt and develop the skills and strategies needed for success [45]. Based on our results and the medical education literature, we recommend the following best practices to deliver any interventional program leading to improved learning and study strategies among medical students: (i) Use a prediction and prevention of failure approach using a validated assessment tool to identify potentially at-risk students. Currently, there are several diagnostic tests available on the market, besides the diagnostic test we used in the present study, such as the Kolb Learning Style Inventory and the Myers-Briggs Type Indicator [11]. Planning an interventional program based on baseline diagnostic scores would allow a more efficient and tailored implementation. (ii) Build a multidisciplinary team to design a robust, thoughtful, and well-grounded intervention program. (iii) Implement the intervention for at least one whole semester and with a minimum of five sessions [41,46]. (iv) Make attendance compulsory in order to avoid loss of follow-up among student participants; evidence shows higher success rates among students attending a course mandatorily versus voluntarily [45,47].

## 5. Conclusions

In summary, using a pre-post study, we found that a tailored interventional program can improve learning and study skills in a cohort of Latin American medical students. Furthermore, this kind of intervention could lower the attrition rate seen among medical schools that receive students without previous college experience.

## Figures and Tables

**Figure 1 healthcare-11-00375-f001:**
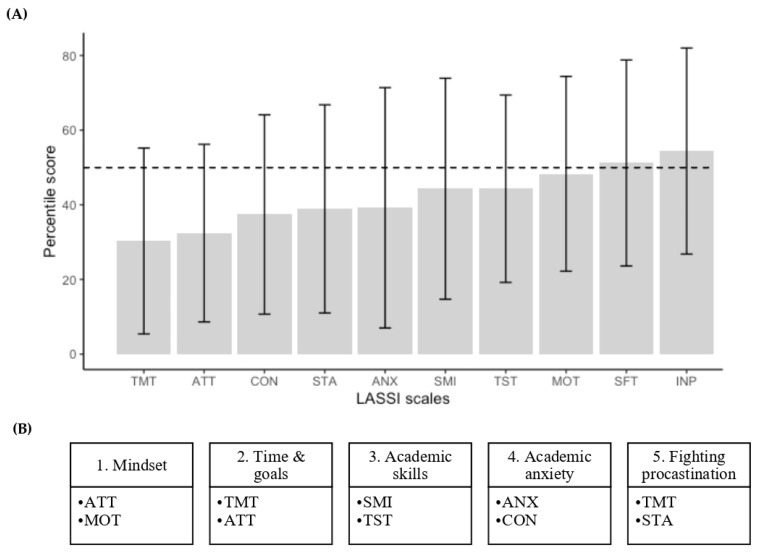
Pre LASSI test scores by scales and planned workshops. Panel (**A**) shows the mean percentile values of the ten LASSI scales before intervention, from the lowest to the highest. The horizontal dashed-black line shows the 50th percentile cut-off value. The error bars represent the standard deviation of the mean value. Panel (**B**) shows the organization of the workshops planned for the intervention phase according to LASSI scales. Abbreviations: INP, Information Processing; SMI, Selecting Main Ideas; TST, Test Strategies; ANX, Anxiety; ATT, Attitude; MOT, Motivation; CON, Concentration; SFT, Self-Testing; STA, Study Aids; and TMT, Time Management.

**Figure 2 healthcare-11-00375-f002:**
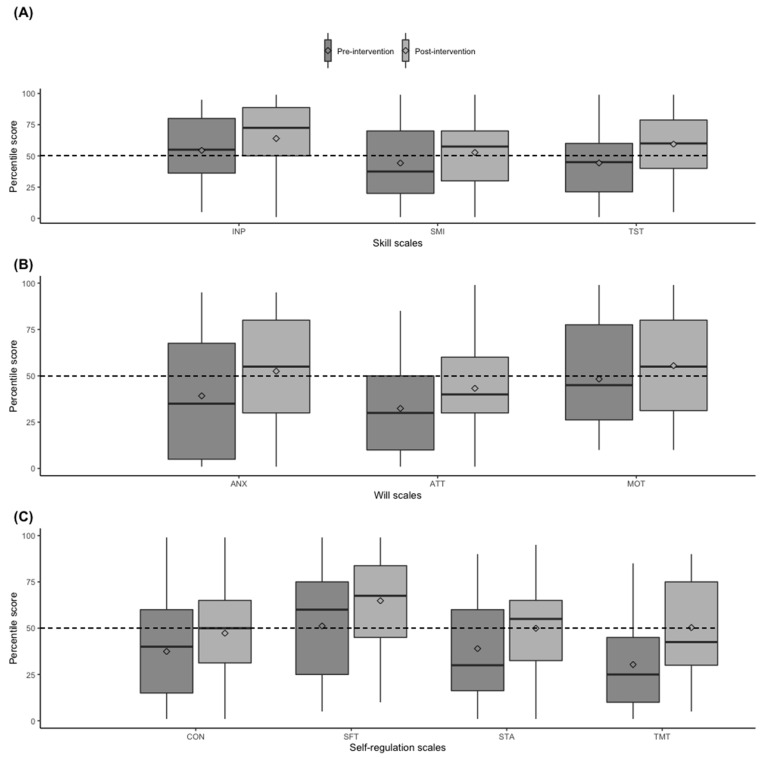
Comparison of pre- and post-LASSI test scores by scale components attained during the USFQ’s Keys to Academic Success Program. Dark gray and light gray boxes correspond to pre- and post-intervention results. Data are shown as median and 25th and 75th percentiles (boxes) and a full range of values (whiskers). Panel (**A**) shows the scales of the Skill component. Panel (**B**) shows the scales of the Will component. Panel (**C**) shows the scales of the Self-regulation component. Abbreviations: INP, Information Processing; SMI, Selecting Main Ideas; TST, Test Strategies; ANX, Anxiety; ATT, Attitude; MOT, Motivation; CON, Concentration; SFT, Self-Testing; STA, Study Aids; TMT, Time Management. Diamond figure depicts the mean value.

**Figure 3 healthcare-11-00375-f003:**
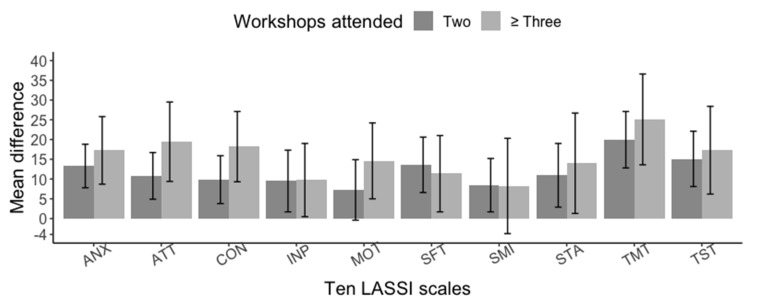
Sensitivity analysis of pre- and post-LASSI test scores by the number of workshops attended during the USFQ’s Keys to Academic Success Program. Abbreviations: ANX, Anxiety; ATT, Attitude; CON, Concentration; INP, Information Processing; MOT, Motivation; SFT, Self-Testing; SMI, Selecting Main Ideas; STA, Study Aids; TMT, Time Management; TST, Test Strategies.

**Table 1 healthcare-11-00375-t001:** Overview of the five workshops implemented for the USFQ’s Keys to Academic Success Program.

		KOLB			
Workshop	LASSI Scales Addressed	Concrete Experience	Reflective Observation	Abstract Conceptualization	Planning Active Experimentation
Mindset	ATT MOT	True vs. False game: Goals, values and motivation Mindset card group exercise	Which group of cards best represents my way of thinking?	- Mindset theory - Motivation: Control locus & learned hopelessness	- Values COMPASS worksheet
2. Time & Goals	TMT ATT	The Jar of Life exercise	Time management: Expectation vs. Reality	- Prioritization - Reflection - SMART Goals	- SMART Goal Template - Goal achievement plan
3. Academic Skills	SMI TST	- Monkey Business illusion video - Brain games	Study habits	- Cognitive functions: attention and inhibition - Identifying main ideas from text - Identifying central ideas - Material interpretation - Synthesis - Study for medical tests	- Main ideas exercise - Memory exercise
4. Academic Anxiety	ANX CON	Documentary fragment: Angst	The Holmes-Rahe Stress Inventory	How do college students experience stress? Data and factors	- Strategies checklist of self-care ideas
5. Overcoming Procrastination	TMT STA	Share study strategies that works for you	How do you study when you don’t feel like it?	- Time management - Procrastination - Intrinsic motivation	Pomodoro Technique

SMART = Specific, measurable, achievable, relevant, and time-bound; ATT = Attitude; MOT = Motivation; TMT = Time Management; ANX = Anxiety; CON = Concentration; INP = Information Processing; SMI = Selecting Main Ideas; STA = Study Aids; SFT = Self-Testing; TST = Test Strategies.

**Table 2 healthcare-11-00375-t002:** Baseline characteristics of the USFQ’s Keys to Academic Success Program participants.

Characteristic	All
	n = 78
Age (yr), mean ± SD	21.4 ± 1.2
Sex, n (%)	
Female	51 (65.4)
Living arrangements, n (%)	
Alone	10 (12.8)
Roommate	11 (14.1)
Family	57 (73.1)
Medical parents, n (%)	
Yes	11 (14.1)
High school attended, n (%)	
Public	7 (9)
Private secular	58 (74.3)
Religious	10 (12.8)
Municipal	3 (3.8)
College degree before Med School, n (%)	
Yes	3 (3.8)
English level perception, n (%)	
Basic	7 (9)
Intermediate	34 (43.6)
Advanced	37 (47.4)
GPA medicine/4, mean ± SD	3.32 ± 0.31
GPA high school/10, mean ± SD	9.3 ± 0.39
Pasteur scholarship, n (%)	
Yes	7 (8.9)
Financial assistantship, n (%)	
Yes	40 (51.3)
Chronic disease, n (%)	
Yes	7 (8.9)
Mental disease, n (%)	
Yes	10 (12.8)
Alcohol, n (%) *	
Yes	46 (58.9)
Permanent drug treatment, n (%)	
Yes	20 (25.6)

Abbreviations: SD, standard deviation; GPA, grade point average. * This variable was assessed with the following question: “In the past year, how often did you drink five or more drinks of any alcoholic beverage or combination of beverages in a single day?”

**Table 3 healthcare-11-00375-t003:** Mean and SD of pre- and post-LASSI ten scales (n = 62).

Scale	Pre-Test	Post-Test	Mean Difference (95% CI)	Percent Change between Pre-Test and Post-Test	*p*-Value *
ANX	39.2 ± 32.2	52.5 ± 28.9	13.3 (7.8, 18.8)	33.9	<0.001
ATT	32.4 ± 23.8	43.3 ± 25.2	10.8 (4.9, 16.7)	33.6	<0.001
CON	37.4 ± 26.7	47.3 ± 25.8	9.8 (3.8, 15.9)	26.5	<0.01
INP	54.4 ± 27.6	63.9 ± 27.5	9.5 (1.7, 17.3)	17.5	0.02
MOT	48.3 ± 26.1	55.5 ± 26.9	7.2 (−0.4, 14.9)	14.9	0.06
SFT	51.2 ± 27.6	64.8 ± 21.7	13.6 (6.6, 20.6)	26.6	<0.001
SMI	44.3 ± 29.6	53.7 ± 26.6	8.5 (1.7, 15.2)	21.2	0.01
STA	38.9 ± 27.9	49.9 ± 26.2	10.9 (2.9, 19)	28.3	<0.01
TMT	30.3 ± 24.9	50.3 ± 27.1	19.9 (12.8, 27.1)	66	<0.001
TST	44.3 ± 25.1	59.4 ± 23.6	15.1 (8.1, 22.1)	34.1	<0.001

Abbreviations: ANX, Anxiety; ATT, Attitude; CON, Concentration; INP, Information Processing; MOT, Motivation; SFT, Self-Testing; SMI, Selecting Main Ideas; STA, Study Aids; TMT, Time Management; TST, Test Strategies. * *p*-values were calculated by using the paired-sample *t*-test.

## Data Availability

Data are available on reasonable request to the corresponding author.

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
