# Peer review of "Improving Learning and Study Strategies in Undergraduate Medical Students: A Pre-Post Study"

_healthcare, 2023, doi:10.3390/healthcare11030375_

Round 1

Reviewer 1 Report

This is a well-researched and prepared manuscript on the important topic of providing strategies to support medical students learning. There are a few errors in grammar and 1 typo as indicated in my copy of the manuscript. 

Additionally, Appendix A Supplementary Table 1 also contains a grammatical error.

Author Response

Response to reviewer 1 comments

Point 1: This is a well-researched and prepared manuscript on the important topic of providing strategies to support medical students learning. There are a few errors in grammar and 1 typo as indicated in my copy of the manuscript.

Response 1: We thank the reviewer for this comment. In response, we have corrected the highlighted errors and typo in the document.

Point 2: Additionally, Appendix A Supplementary Table 1 also contains a grammatical error.

Response 2: We thank the reviewer for pointing this out this error. In response we have correct this error in the supplementary document.

Reviewer 2 Report

1. add background to the abstract

2. add space before reference 6 in the introduction

3. add all the abbreviations immediately under the figure 1

5. page 5 add few enters as it is hard to read

In general, this is an interesting and well-conducted study for all medical educators. References, tables and figures are well presented. I believe this study will add to the body of literature in this medical field

Author Response

Response to Reviewer 2 Comments

Point 1: add background to the abstract

Response 1: We thank the reviewer for this comment. However, currently the abstract has the allowed 200 words.

Point 2: add space before reference 6 in the introduction

Response 2: We appreciate the reviewer´s comment. In response, we have added the recommended space.

Point 3:  add all the abbreviations immediately under the figure 1

Response 3: We thank the reviewer for this comment. I do confirm that the necesary abbreviations were present in the initial submittted manuscript. However, after the original manuscript was transfer to the journal´s template it moved the abbreviations text. In response to this comment, we have edited the text accordingly.

Point 4: page 5 add few enters as it is hard to read

Response 4: We thank the reviewer for this comment. In response, we have added the necessary enters to ease the reading of this section.

Reviewer 3 Report

The manuscript reports about a pre-post intervention to support learning and study strategies in undergraduate medical students. Within the limitations of a pre-post study, the study is well done and the text is clear. I checked it against the DoCTRINE reporting guideline (https://www.equator-network.org/reporting-guidelines/the-doctrine-guidelines-defined-criteria-to-report-innovations-in-education/) and I found the manuscript compliant. A first suggestion is hence to quote DoCTRINE in the text .

I have few more suggestion to improve the manuscript:

- the overall tackled problem is the "struggling student". I suggest that in the Intro the authors could add few lines to frame the problem of study skill in the more general problem of students' dropout. Study skill is one of the reasons for the dropout, together with health (mental health!!) and socio-economic problems

- page 3, par. 2.4 "The inventory is a self-report instrument with 60 items divided into 10-scales": how are the items graded? Likert? how many grades?

- page 8, bottom "After the workshop interventions, only the ATT and CON scales did not surpass the 50th percentile cutoff value". As far as I can see in Figure 2 and Table 3 also TMT remained under the threshold, although it had the highest % increase. The same misunderstanding seems to come also at page 10 "ATT and CON scales showed statistical difference in the post LASSI scores, they did not surpass the 50th percentile threshold " 

- Limitations
-- page 11, top "The pre-post study design and sensitivity analysis performed confer evidence to suggest causality of the implemented interventional program and its impact." You could make this statement only with a comparative study. There are other factors that could explain the effect, starting from only having being exposed to the LASSI. I suggest that you coud change "causality" (a very strong and "slippery" concept) with something like "a role (component, factor, ...) in the observed effect"

-- page 10, few lines below "However, we found no differences between both groups in demographic variables so we anticipate a low probability of selection bias" This statement only implies that demo variables were not responsible. What about the score at the pre-test of those who did not sit the post-test? MOT? ANX? ATT?

A final remark, that is a true question as an interested reader of your manuscript. I noticed that the mean score increased, but the variability as expressed by the boxes and - mainly - the whiskers did not seem to change that much. Can we hypothesize that the students who scored highest and lowest are two distinct groups and that those who had the lowest scores are the group at high risk of dropout?    

Author Response

Response to Reviewer 3 Comments

Point 1: The manuscript reports about a pre-post intervention to support learning and study strategies in undergraduate medical students. Within the limitations of a pre-post study, the study is well done and the text is clear. I checked it against the DoCTRINE reporting guideline (https://www.equator-network.org/reporting-guidelines/the-doctrine-guidelines-defined-criteria-to-report-innovations-in-education/) and I found the manuscript compliant. A first suggestion is hence to quote DoCTRINE in the text .

Response 1: We thank the reviewer for this comment. In response, we have added the following text at the end of the Procedure section of the manuscript: "The DoCTRINE (Defined Criteria To Report INnovations in Education) guideline was used to ensure the reporting of this educational innovation study (33)."

Point 2: the overall tackled problem is the "struggling student". I suggest that in the Intro the authors could add few lines to frame the problem of study skill in the more general problem of students' dropout. Study skill is one of the reasons for the dropout, together with health (mental health!!) and socio-economic problems

Response 2: We appreciate the reviewer´s comment. In response to this comment, we have added the following thext in the introduction section: "Some factors associated with medical school dropout includes socio-demographic problems, lower entry qualifications, psychological attributes and signs of academic struggle during medical school coursework. Being this last factor strongly associated with dropout (7)."

Point 3: - page 3, par. 2.4 "The inventory is a self-report instrument with 60 items divided into 10-scales": how are the items graded? Likert? how many grades?

Response 3: We thank the reviewer´s comment. Yes, the LASSI test to grade each item uses a 5-point Likert scale. In response to this comment, we have added this information in the "Measures" section of the manuscript.

Point 4: - page 8, bottom "After the workshop interventions, only the ATT and CON scales did not surpass the 50th percentile cutoff value". As far as I can see in Figure 2 and Table 3 also TMT remained under the threshold, although it had the highest % increase. The same misunderstanding seems to come also at page 10 "ATT and CON scales showed statistical difference in the post LASSI scores, they did not surpass the 50th percentile threshold " 

Response 4: We thank the reviewer for this comment and the opportunity to clarify this misunderstanding. Figure 2 shows median values and Table 3 shows mean values. In response to this comment, we have added diamond figures to Figure 2 to depict the mean value and also edited text in the results and discussion sections as appropriate to clarify this issue.

Point 5: -- page 11, top "The pre-post study design and sensitivity analysis performed confer evidence to suggest causality of the implemented interventional program and its impact." You could make this statement only with a comparative study. There are other factors that could explain the effect, starting from only having being exposed to the LASSI. I suggest that you coud change "causality" (a very strong and "slippery" concept) with something like "a role (component, factor, ...) in the observed effect"

Response 5: We thank the reviewer for pointing this out and we agree. In response to this comment, we have edited this section as suggested.

Point 6: -- page 10, few lines below "However, we found no differences between both groups in demographic variables so we anticipate a low probability of selection bias" This statement only implies that demo variables were not responsible. What about the score at the pre-test of those who did not sit the post-test? MOT? ANX? ATT?

Response 6: We appreciate the reviewer´s comment. In response to this comment, we built a table to compare the pre-test scores between those who did and did not sit the post-test. This new table (Supplementary Table 2) has been added in the supplementary material file. Also, we acknowledged this limitation in the discussion section of the manuscript.

Point 7: A final remark, that is a true question as an interested reader of your manuscript. I noticed that the mean score increased, but the variability as expressed by the boxes and - mainly - the whiskers did not seem to change that much. Can we hypothesize that the students who scored highest and lowest are two distinct groups and that those who had the lowest scores are the group at high risk of dropout?  

Response 7: We thank the reviewer for this comment. Based on our results we can support the hypothesis that students who scored highest and lowest are different groups. However, we also did find that students with the lowest pre LASSI scores overall attended more workshops and those students were more likely to obtained a higher percent change improvement in the post LASSI test. Thus, we cannot support the hypothesis that those students (with lowest scores) is the group at high risk of dropout.